# Therapeutic Drug Monitoring of Quinidine in Pediatric Patients with *KCNT1* Genetic Variants

**DOI:** 10.3390/pharmaceutics14102230

**Published:** 2022-10-19

**Authors:** Alessandro Ferretti, Raffaele Simeoli, Sara Cairoli, Nicola Pietrafusa, Marina Trivisano, Carlo Dionisi Vici, Nicola Specchio, Bianca Maria Goffredo

**Affiliations:** 1Rare and Complex Epilepsy Unit, Department of Neuroscience, Bambino Gesù Children’s Hospital, IRCCS, 00165 Rome, Italy; 2Department of Pediatric Specialties and Liver-Kidney Transplantation, Division of Metabolic Diseases and Drug Biology, Bambino Gesù Children’s Hospital, IRCCS, 00165 Rome, Italy

**Keywords:** therapeutic drug monitoring (TDM), quinidine, pediatric patients, KCNT1, DEE, seizures, anti-seizure medications (ASMs)

## Abstract

Quinidine (QND) is an old antimalarial drug that was used in the early 20th century as an antiarrhythmic agent. Currently, QND is receiving attention for its use in epilepsy of infancy with migrating focal seizures (EIMFS) due to potassium sodium-activated channel subfamily T member 1 (*KCNT1*) genetic variants. Here, we report the application of Therapeutic Drug Monitoring (TDM) in pediatric patients carrying *KCNT1* genetic variants and orally treated with QND for developmental and epileptic encephalopathies (DEE). We measured plasma levels of QND and its metabolite hydroquinidine (H-QND) by using a validated method based on liquid chromatography coupled with mass spectrometry (LC-MS/MS). Three pediatric patients (median age 4.125 years, IQR 2.375–4.125) received increasing doses of QND. Cardiac toxicity was monitored at every dose change. Reduction in seizure frequency ranged from 50 to 90%. Our results show that QND is a promising drug for pediatric patients with DEE due to *KCNT1* genetic variants. Although QND blood levels were significantly lower than the therapeutic range as an anti-arrhythmic drug, patients showed a significant improvement in seizure burden. These data underlie the utility of TDM for QND not only to monitor its toxic effects but also to evaluate possible drug–drug interactions.

## 1. Introduction

Quinidine (QND) is an old antimalarial and antiarrhythmic drug that was used in the early 20th century as an antiarrhythmic agent to maintain sinus rhythm after the conversion from atrial flutter or atrial fibrillation and to prevent the recurrence of ventricular tachycardia or ventricular fibrillation [1]. The use of QND has often been associated with an increased risk of ventricular arrhythmia and sudden death, together with numerous adverse effects and drug interactions [1]. Therefore, QND has been withdrawn from its use and, in recent years, has become unavailable in many countries [1].

Due to its ability to inhibit the sodium-activated potassium channel encoded by a potassium sodium-activated channel subfamily T member 1 (*KCNT1*) gene, QND has been used in epilepsy of infancy with migrating focal seizures (EIMFS) (#608167) caused by *KCNT1* genetic variants [2]. EIMFS is a rare epileptic syndrome characterized by epilepsy onset in the first six months of life associated with severe developmental delay, severe disability, resistance to conventional anti-seizure medications (ASMs), and poor prognosis [3].

EIMFS pathophysiology involves different molecular processes: gene and protein regulation, ion channel function, and solute trafficking. Recently, up to 33 genes have been related to this condition [4].

The literature data identify *KCNT1* as the major player involved [4,5]. Furthermore, genetic variants in *KCNT1* have emerged as an important cause of epilepsy with a wide phenotypic spectrum, including autosomal dominant nocturnal frontal lobe epilepsy (ADNFLE) in children and adults and an early infantile developmental and epileptic encephalopathy (EIDEE) in infants and children [6].

*KCNT1* encodes for potassium (K^+^) channel subunits that share similar structural arrangements with the classical voltage-gated K_v_ channel subunits [7]. In vitro studies reported that *KCNT1* genetic variants are responsible for higher voltage currents compared to wild-type *KCNT1* channels in EIMFS; these results suggest a gain of function mechanism underlying the epileptogenesis associated with *KCNT1* genetic variants [5]. This gain of function may be reversed by QND.

Indeed, target therapy with QND has been introduced for treating patients with EIMFS with variable results, ranging from successful [8] to poor response or toxicity [9,10,11].

Beyond QND, in two observational studies of a patient with KCNT1-related epilepsy, data revealed that multiple ASMs, first of all, ketogenic diet, vigabatrin, and cannabidiol, may be helpful in reducing seizure frequency in such patients, though it could not be established a clear-cut superiority between these treatments [12,13].

Therapeutic drug monitoring (TDM) could be a useful strategy to optimize therapy with QND when administered in association with other ASMs in pediatric patients. This approach allows a more tailored pharmacological management reducing risks of sub-therapeutic drug exposure or toxicity.

In this manuscript, we describe three cases of pediatric patients carrying *KCNT1* genetic variants treated with QND. TDM of QND and its active metabolite hydroquinidine (H-QND) was performed to better optimize the concomitant ASMs and to avoid the appearance of adverse drug reactions (ADRs).

## 2. Materials and Methods

### 2.1. Study Design and Patients’ Characteristics

This is a retrospective study evaluating three patients with an EIMFS related to *KCNT1* genetic variants who received QND from May 2019 to May 2022 at the Rare and Complex Epilepsy Unit of the Bambino Gesù Children’s Hospital in Rome, Italy. QND was started in each of the patients at a dose of 5 mg/kg/day TID, with increases of 3–5 mg/kg/day every 1–2 weeks based on the family’s ability to follow clinical and cardiological evaluations. QND galenic formulation in oral suspension was prescribed to each patient and purchased at the pharmacy in their area of residence. Clinical efficacy (frequency of seizures) and adverse events were recorded.

TDM for quinidine was applied as routine clinical practice in the three patients. The Ethical Review Board of our Hospital was informed about this TDM application, but no formal protocol was submitted. Plasma samples were collected at least 4 days following the start of QND administration to measure drug levels at a steady state. Whole blood was collected from an indwelling arterial line in EDTA tubes and centrifuged at 3500× *g* for 5 min to obtain plasma.

All patients were subjected to close cardiological monitoring with ECG and QTc evaluation during QND treatment. These examinations were normal except when specified. To measure QND concentration and monitor cardiac toxicity, we sent plasma samples to the TDM lab for analysis of drug levels at the steady state. During QND titration, cardiac activity and drug plasma levels were monitored at each dose increase. Thereafter, during follow-up and in the absence of dose changes, TDM was applied less frequently.

### 2.2. Determination of Quinidine and Hydroquinidine Levels by LC-MS/MS

QND and H-QND plasma levels were measured at the Laboratory of Metabolic Diseases and Drug Biology at Bambino Gesù Children’s Hospital in Rome. Liquid chromatography and mass spectrometry analyses were performed by using a UHPLC Agilent 1290 Infinity II 6470 (Agilent Technologies, Santa Clara, CA, USA) equipped with an ESI-JET-STREAM source operating in the positive ion (ESI+) mode. The software used for controlling this equipment and analyzing data was MassHunter Workstation (Agilent Technologies). The assay calibration curves were linear and ranged from 0.025 to 7.60 µg/mL for QND and from 0.025 to 12.60 for H-QND. Each batch of patients’ analyses included both low- and high-quality controls (QCs) at fixed concentrations. For QND, L-QC and H-QC were 0.82 and 5.12 µg/mL, respectively. For H-QND, L-QC and H-QC were 1.15 and 8.65 µg/mL, respectively. Calibrators, QCs, and samples were analyzed using a validated LC-MS/MS kit (MassTox^®^ TDM Anti-arrhythmic drugs) provided by Chromsystems (Chromsystems Instruments & Chemicals GmbH, Gräfelfing, Germany).

This kit included calibrators and QCs, and was validated according to EMA guidelines for bioanalytical methods validation:

(http://www.ema.europa.eu/docs/en_GB/document_library/Scientific_guideline/2011/08/WC500109686.pdf, 21 July 2011).

### 2.3. Statistical Analysis

All graphs were performed using Graph-Pad Prism 7.0 (Graph-Pad Software Inc., San Diego, CA, USA). For this study, no formal power calculation was made.

## 3. Results

**Patient 1** was a male aged 5 years and 9 months; he developed epileptic spasms and focal tonic seizures characterized by asymmetric tonic posturing with eye deviation at the age of 9 months. Seizures were unresponsive to conventional ASMs (ACTH, vigabatrin, phenobarbital, clonazepam, nitrazepam, and topiramate). At the age of 2 years, the EEG showed multifocal ictal activity, shifting from one cerebral region or hemisphere to the contralateral. The electroclinical features of migrating seizures, concomitant with major developmental delay, confirmed the diagnosis of EIMFS.

Whole-exome sequencing revealed a de novo heterozygous mutation in *KCNT1* (c.337G>A, p.Val113Met), determined to be a variant of uncertain significance (VUS) according to ACMG Guidelines [14]. Cardiological evaluation (electrocardiogram ECG, echocardiography) revealed no abnormalities; therefore, at the age of 3 years and 2 months, QND was administered at a starting dose of 5 mg/Kg/day (divided into three doses), without clear-cut improvement in seizure frequency (about 30–40 tonic seizures per day, lasting 1 min each). Concomitant ASMs were topiramate and nitrazepam. The dose of quinidine was increased gradually. At the dose of 12 mg/kg/day, despite seizures remaining unchanged in terms of frequency, we recorded a reduction in their intensity and a slight improvement in participation. First measurements of QND (12 mg/Kg/day) resulted in plasma levels of 0.133 μg/mL (Figure 1a).

At the age of 3 years and 4 months, we increased the QND dose up to 15 mg/kg/day with a concomitant 60% reduction in seizures’ frequency and intensity (seizures dropped down from 30 to 10 episodes per day, each lasting less than 30 s).

At the QND dose of 15 mg/kg/day, a slight prolongation of QTc interval (470 msec) in the ECG was seen, and the dose was reduced to 12 mg/kg/day until QTc normalization. Concomitant ASMs were unchanged. Dose reduction resulted in a QND plasma concentration of 0.073 μg/mL. Normalization of QT allowed us to slowly increase the QND dose from 12 up to the current dose of 40 mg/kg/day. At the same time, we started to also monitor H-QND levels. Upon reaching the current QND dose, seizure frequency decreased by approximately 90% (less than 3–5 episodes per day). A QND dose of 17.5 mg/kg/day led to a QND and H-QND plasma concentration of 0.096 and 0.045 μg/mL, respectively. At the last follow-up visit, this patient was receiving 40 mg/kg/day of QND resulting in plasma levels of 0.80 and 0.073 μg/mL for QND and H-QND, respectively. EEG performed at the age of 5.5 years showed slowing of the background activity with bitemporal spikes and an absence of physiological sleep figures. During the follow-up, interictal epileptiform abnormalities decreased, but no improvement in the background was seen. The brain MRI at 6 months of age was reported as normal except for a thickness of corpus callosum; a second brain MRI at the age of 2.5 years showed progressive supratentorial and cerebellar atrophy (Figure 2). At the last follow-up (5 years and 9 months of age), cognitive and motor development was severely delayed; neurological examination revealed hand apraxia, poor eye contact with minimal response to social interactions, and severe hypotonia: these symptoms minimally improved with QND.

**Patient 2** was a 2 year and 3 months old male; at 2 days old, he started to present with asymmetrical tonic seizures persisting over time despite multiple ASMs. From the age of 3 months, seizure semiology changed: focal clonic, asymmetrical epileptic spasms, and generalized tonic seizures with eyelid myoclonia appeared. At the age of 4 months, EEG showed a clear-cut migrating seizure pattern, which allowed the diagnosis of EIMFS (Figure 3). Seizures were refractory to several ASMs (phenobarbital, pyridoxine, biotin, folinic acid, phenytoin, levetiracetam, carbamazepine, clonazepam, vigabatrin, potassium bromide). Whole-exome sequencing revealed a de novo heterozygous mutation in *KCNT1* (c.1283G>A, p.Arg428Gln) and in *GRIN2B* (c.448A>G, p.Ile150Val), determined to be a pathogenic and likely pathogenic according to ACMG Guidelines [14], respectively. QND 5 mg/Kg/day was started at the age of 7 months. Concomitant ASMs were phenobarbital, clonazepam, vigabatrin, and potassium bromide. The first measurement of QND and H-QND blood levels, performed at a dose of 20 mg/Kg/day, were 0.15 and 0.050 μg/mL, respectively (Figure 1b).

When we increased QND dosage up to 40 mg/kg/day, resulting in a QND and H-QND plasma concentration of 0.18 μg/mL and 0.050 μg/mL, respectively, seizure frequency decreased by approximately 20%.

From the age of 8 months, seizure frequency worsened; he also presented a refractory convulsive status epilepticus (SE) treated with i.v. Midazolam continuous infusion. An increase in QND up to 60 mg/kg/day resulted in a 50% reduction in seizure frequency with a plasma concentration of 0.26 μg/mL.

With a prescription of 60 mg/kg/day, he presented weekly, isolated focal tonic and migrating seizures lasting 20–30 s. At the last follow-up visit, he was receiving 60 mg/kg/day. QND and H-QND plasma levels were 0.23 and 0.040 μg/mL, respectively. EEG recordings showed a slow background activity with poor differentiation between wakefulness and sleep, together with recurrent slow waves intermingled with epileptiform abnormalities, predominantly over the posterior regions. No EEG modifications were recorded during QND treatment. Brain MRI performed at the age of 1 month was normal. At the last follow-up (2 years and 3 months of age), psychomotor development was severely delayed, with no responses to social interactions, language was not acquired, and severe hypotonia was evident. A slight improvement in eye contact was observed with QND therapy.

**Patient 3** is a male aged 2 years and 6 months. He started to present with focal tonic seizures when he was 40 days old. Conventional ASMs (valproate, clobazam, phenytoin, levetiracetam, carbamazepine) and the ketogenic diet were ineffective. The frequency of focal tonic and focal clonic seizures gradually increased to 20 times daily at the age of 11 months. Whole-exome sequencing revealed a de novo heterozygous genetic variant in *KCNT1* (c.862G>A, p.Gly288Ser), determined to be pathogenic according to ACMG Guidelines [14]. At the age of 11 months, he started QND at 5 mg/kg/day in combination with phenobarbital and carbamazepine. At 15 mg/kg/day, plasma QND and H-QND concentrations were 0.007 and 0.004 μg/mL, respectively (Figure 1c). Seizure frequency decreased by approximately 75% upon reaching the QND dose of 40 mg/kg/day, corresponding to QND and H-QND plasma levels of 0.020 and 0.011 μg/mL, respectively.

At the last follow-up visit, he was still receiving 40 mg/kg/day, and QND plasma levels were 0.09 μg/mL. EEG revealed moderately slow background activity with epileptiform abnormalities over the bilateral frontal regions increasing during sleep. Soon after QND administration, there was a remission of epileptiform abnormalities and a better organization of brain electrical activity. The brain MRI performed at 2 months old was reported as normal. At the age of 2 years and 6 months, his psychomotor development was delayed with poor interaction and absence of language; he had a spontaneous poor motor initiative in the context of diffuse hypotonia. No improvement in psychomotor development was observed with QND therapy.

## 4. Discussion

Based on previous reports and our clinical experience, QND could be considered a promising drug for patients with DEE due to *KCNT1* genetic variants [2,8,11,12]. Its use is still debated due to the limited number of cases, the poor knowledge of the therapeutic range, and the appropriate titration methods. Similarly, studies published so far reported contrasting results [9,10,12,15].

In a large cohort of patients with KCNT1-related epilepsy, 20% of subjects receiving QND treatment showed a >50% reduction in seizure frequency, with sustained seizure freedom occurring only in one patient [12]. QND responsiveness could change over time, and patients reporting initial seizure freedom may experience a recurrence of seizures [12]. Patients who responded best to QND carried variants located immediately distal to the NASD domain within the RCK2 domain [12], the region of the protein hypothesized to be important in coupling sensitivity to intracellular sodium levels with channel gating [16,17]. None of the genetic variants of our patients are included in the latter domain. In the study of Fitzgerald et al., therapeutic blood levels were not achieved in all patients, with 45% of subjects failing to reach a QND blood level of 2 μg/mL [12].

To date, the only controlled trial on QND has been performed in adult patients with *KCNT1* genetic variants [15]; this trial demonstrated that QND was not effective in reducing seizure burden and that QTc prolongation was a significant, dose-limiting side effect. However, in this trial, there were some limitations, including the small sample size and the failure to achieve therapeutic serum levels due to the appearance of side effects [13]. In a case report on two pediatric patients affected by drug-resistant epilepsy caused by *KCNT1* mutations and treated with QND, only one showed an 80% reduction in seizure frequency, while the second patient did not improve at all [18].

In contrast, all our patients showed benefits from QND therapy, with a significant reduction in seizure frequency ranging from 40 to 90%. These positive results may be explained by the younger age of our patients if compared with the previous studies.

Moreover, as reported in other cases [18,19], although psychomotor development remains severely delayed in all patients, we observed a slight improvement with an increase in eye contact in two of our patients.

Neuroimaging has not been analyzed systematically in KCNT1-related DEE, although scattered case reports have documented either normal brain anatomy or—less often—cortical atrophy and thin corpus callosum [20,21]. In our three patients, the brain MRI performed at the epilepsy onset was normal, and in patient 1, in whom the neuro-imaging was repeated at 2.5 years, there was cerebral and cerebellar atrophy with a thinning of the corpus callosum.

In agreement with the emerging scientific literature, in our patient, the relationship between QND dose and plasma concentration was inconsistent, particularly at high doses of QND [11,12].

As previously published [12], in our cases, QND plasma levels were significantly lower if compared to the suggested therapeutic range (1–5 μg/mL) [22,23].

Nevertheless, we observed an improvement in seizure burden. This peculiar finding could be partially explained by the complex pharmacokinetics properties of QND, which could also explain the high inter-individual variability [24]. In fact, the absorption rate following oral administration could be affected by the type of oral formulation. Moreover, QND metabolism produces different pharmacologically active compounds with an impaired capability of protein binding. Therefore, even though total plasma concentrations of QND metabolic products are lower than intact QND, their lower protein binding could indicate that a larger part of total metabolite concentration is in unbound form and, therefore, pharmacologically active [24]. Moreover, QND metabolites may significantly contribute to both therapeutic and toxic effects observed during therapy with QND. For example, the cardiac effects of hydroquinidine (H-QND) have already been evaluated in patients with Brugada syndrome [25]. Here, we measured hydroquinidine levels alongside QND plasma concentrations. H-QND were lower than QND levels; however, in the absence of specific therapeutic ranges and previous studies on the anti-epileptic effects of H-QND, we were not able to draw exhaustive conclusions. It is worth considering the contribution of renal excretion and hepatic metabolism in the elimination of QND. For the first process, the younger age of our patients could allow faster renal elimination compared to older subjects. For the second aspect, it is worth considering the contribution of concomitant medications that could induce QND metabolism. QND has also been shown to produce drug–drug interactions (DDIs) with other ASMs [24]. Drug-disposition studies of orally administered QND in healthy volunteers indicated that phenobarbital and phenytoin, via the induction of CYP3A4, reduced the half-life by approximately 50% and, therefore, steady-state plasma concentration [11,26,27]. Careful QND titration should be planned when drugs that induce CYP3A4 are used in combination [11]. It is not known if other interactions; however, it is predictable that all inducers reduce, and all inhibitors increase the QND blood levels. In this context, TDM of QND and H-QND may help to optimize dosing regimens. In our report, two out of three patients were receiving phenobarbital as concomitant ASM. Therefore, both QND and H-QND plasma concentrations were monitored not only to prevent quinidine-related cardiotoxicity but also to evaluate the impact of phenobarbital co-administration on QND circulating levels.

The main QND-related side effect is QT elongation on the ECG; however, sedation, arrhythmia, elevated liver function tests, and rash (5%) were also reported [8,11,12,18,28,29,30,31].

Our Patient 1 reported a QT elongation despite a low dose of QND (15 mg/kg/day) and low plasma concentration (0.27 μg/mL); similar cases were reported by Abdelnour E et al. [28] with QT elongation in serum quinidine concentration of 0.40 μg/mL and by Mikati MA et al. [18] at a low dose of 34.4 mg/kg/day.

These data demonstrate that achieving a truly safe range of serum QND levels at appropriate doses is not easily reliable since the occurrence of adverse reactions may depend on several aspects, also including genetic factors and the electrolytic balance. However, it is well established that a QND dose over 74.5 mg/kg/day or a serum QND concentration above 9.4 μg/mL is responsible for serious complications [11].

Considering the well-documented QND cardiotoxicity, the effects of concomitant medications that induce QND metabolism, reducing its serum levels, and the lack of defined titration schemes for treating patients with EIMFS, more studies could be useful to investigate the effects of other concomitant ASMs on QND plasma levels. Furthermore, the concomitant use of drugs that are known to cause prolongation of QT interval should be avoided if possible [32].

Based on these considerations, QND is potentially one of the few drugs available for treating patients with *KCNT1* mutations and EIMFS, TDM could be advisable to guarantee the safe and effective use of this drug.

However, it is worth mentioning that our report has some limitations. First, this was a small and retrospective study conducted on three patients. Additionally, pharmacological treatment responsiveness was evaluated by measuring the reduction in clinically evident seizures as reported by caregivers and clinicians, and it may have been conditioned by the placebo effect due to the open-label nature of the treatment. In contrast, an electroencephalographic seizure burden assessment may be more reliable in measuring treatment responsiveness. Finally, we are aware that our is only a small observational report and that proper clinical studies will be necessary to explore the PK/PD properties and establish an appropriate therapeutic range for both QND and H-QND in pediatric patients with *KCNT1* genetic variants and EIMFS. However, in the absence of specific guidelines, we believe that TDM of QND and its metabolite could be a valid tool for future investigations aiming to establish a specific therapeutic range and to allow the safe handling of this drug.

## Figures and Tables

**Figure 1 pharmaceutics-14-02230-f001:**
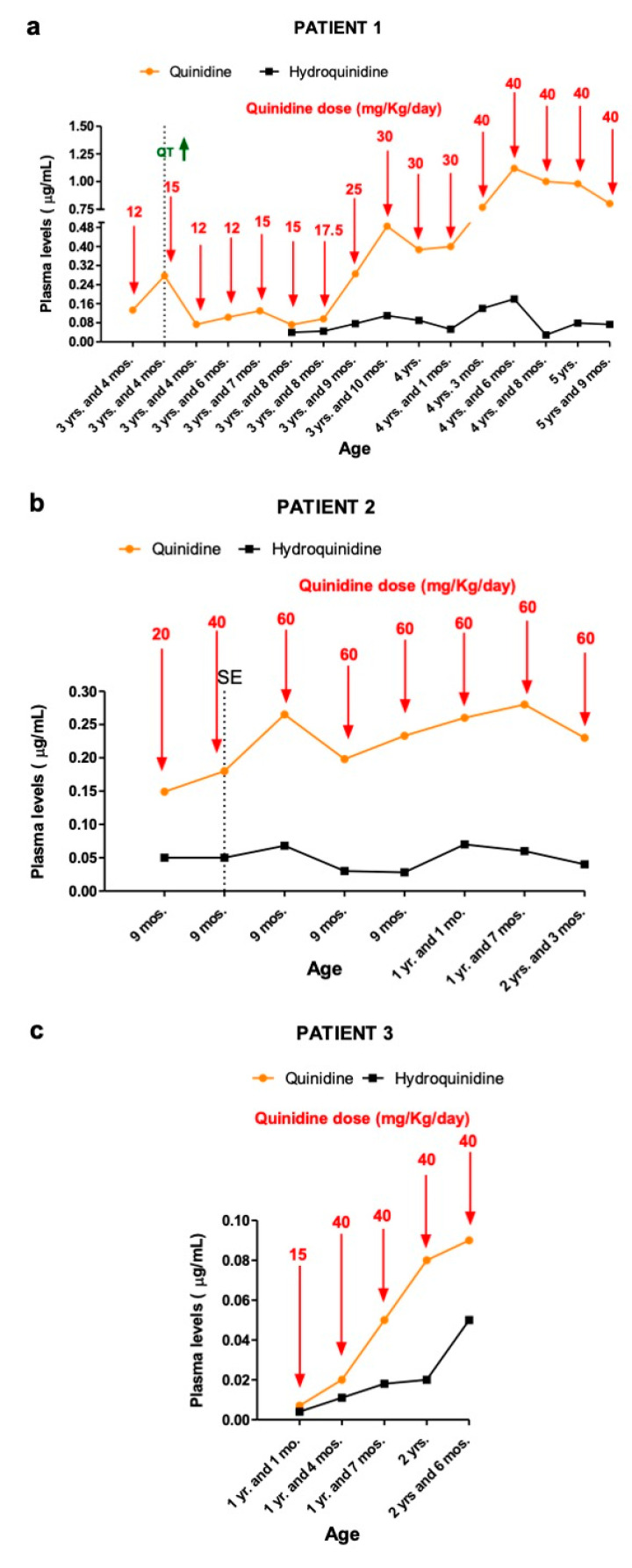
Quinidine and hydroquinidine plasma concentrations in Patient 1 (**a**), Patient 2 (**b**) and Patient 3 (**c**). Abbreviations: SE = Status Epilepticus; yr. = year; yrs. = years; mo. = month; mos. = months. The green arrow indicates the prolongation of QTc interval in the electrocardiogram.

**Figure 2 pharmaceutics-14-02230-f002:**
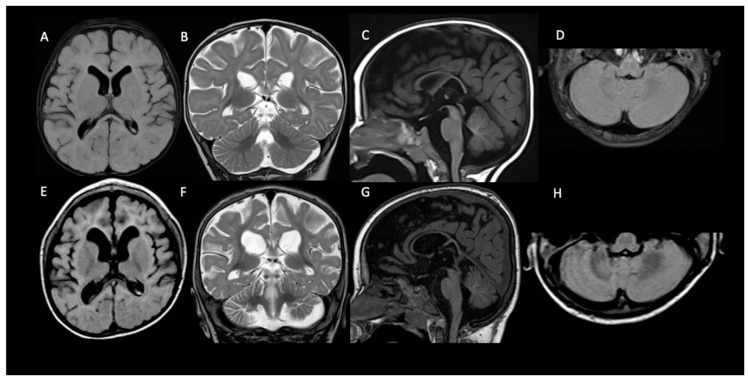
Progressive brain atrophy in patient 1. Compared to brain MRI performed at 6 months of age (**A**–**D**), MRI performed at 2.5 years of age (**E**–**H**) show progressive ventricular and CSF enlargement (**E**,**F**) more prominent in the frontal-temporal regions (**E**) and a widening of the interfolial cerebellar spaces of both cerebellar hemispheres (**G**,**H**) with a further thinning of the corpus callosum (**G**).

**Figure 3 pharmaceutics-14-02230-f003:**
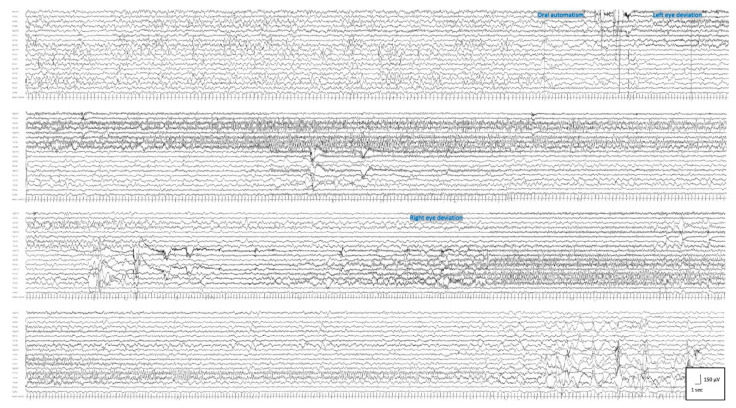
Ictal EEG pattern in Patient #2 at the age of 4 months: seizure started with a flattening in the right fronto-central region, spreading throughout the right hemisphere and migrating to the left hemisphere after around 150 s. The ictal discharge ends in left fronto-temporal region 360 s after its beginning. Clinically, this seizure occurred while awake and is characterized by oral automatisms followed by left eye deviation after 90 s followed by right eye deviation 240 s after its beginning, as shown in the figure. Filters: high pass, 0.01 Hz; and low pass, 35 Hz.

## Data Availability

Data are available on reasonable request from the corresponding author.

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
