# Peer review of "Therapeutic Drug Monitoring of Quinidine in Pediatric Patients with KCNT1 Genetic Variants"

_pharmaceutics, 2022, doi:10.3390/pharmaceutics14102230_

Round 1

Reviewer 1 Report

Ferretti et al., 2022 reported the treatment response of three children with KCNT1-related disorders to quinidine. The authors described the plasma levels and cardiac toxicity of QND and its metabolite H-QND.

The article should be reviewed by a native English speaker because it contains numerous typos.

The authors should include the ACMG criteria for each described mutation (pathogenic, likely pathogenic, VOUS, etc.)

The authors should include the disorder's OMIM reference number when it is first mentioned.

In addition, the authors should describe any known interactions between ASMs and QND. Is phenobarbital the only relationship they wish to discuss? Likewise, they should list all ASDs that can prolong the QT interval in the discussion section.

The authors should note that there was no improvement in psychomotor development for patient 3 receiving QND treatment.

Finally, it is a very interesting article that describes a promising treatment for a rare disorder.

Reviewer 2 Report

Ferretti A. et al. is a well written manuscript that describes therapeutic drug monitoring of quinidine and its metabolite (hydro-quinidine) as used for 3 cases of pediatric patients with KCNT1 genetic variants that were treated with quinidine for the management of epilepsy of infancy with migrating focus seizures (EIMFS).

My comments are as follows:

1.      Fitzgerald et al. (reference#12) found that more favorable response to quinidine was seen in specific KCNT1 variants. I suggest discussing how the KCNT1 variants for the 3 patients included in this study relate to that.

2.      I suggest the authors to discuss the currently used treatment options for EIMFS due to KCNT1 mutation and their limitations

3.      I recommend adding ‘chances of placebo effect due to open label nature of treatment’ as a potential limitation of this study

Other Minor comments:

1.      Page 1 line 21-22 (abstract), please use a decimal point instead of comma for summarizing the age. Same comment for Page 4, line 161.

2.      Page 4 Line 154,  please use ‘resulted’ in place of ‘reflected’

3.      Page 5 line 188, please change ‘ha’ to ‘he’

4.      Page 5 line 204, remove ‘was’

5.      Page 7 line 281, please update to ‘it is worth considering the contribution…’

6.      Page 7 line 302, please change ‘of’ to ‘for’.

7.      Page 7 line 307, please remove ‘considered that’ from the sentence.
